# MACHINE READING COMPREHENSION WITH ENHANCED LINGUISTIC VERIFIERS

## ABSTRACT

We propose two linguistic verifiers for span-extraction style machine reading comprehension to respectively tackle two challenges: how to evaluate the syntactic completeness of predicted answers and how to utilize the rich context of long documents. Our first verifier rewrites a question through replacing its interrogatives by the predicted answer phrases and then builds a cross-attention scorer between the rewritten question and the segment, so that the answer candidates are scored in a *position-sensitive* context. Our second verifier builds a hierarchical attention network to represent segments in a passage where neighbour segments in long passages are *recurrently connected* and can contribute to current segment-question pair's inference for answerablility classification and boundary determination. We then combine these two verifiers together into a pipeline and apply it to SQuAD2.0, NewsQA and TriviaQA benchmark sets. Our pipeline achieves significantly better improvements of both exact matching and F1 scores than state-of-the-art baselines.

## 1 INTRODUCTION

Teaching a machine to read and comprehend large-scale textual documents is a promising and long-standing goal of natural language understanding. This field, so called machine reading comprehension (MRC) (Zhang et al., 2019; 2020c), has achieved impressive milestones in recent years thanks to the releasing of large-scale benchmark datasets and pretrained contextualized language models (CLM). For example, for the well-testified span-extraction style SQuAD2.0 dataset[1], current best results under the framework of pretraining+fine-tuning employing ALBERT (Lan et al., 2020) are 90.7% of exact matching (EM) and 93.0% of F1 score, exceeds their human-level scores of 86.8% and 89.5% (Rajpurkar et al., 2016; 2018) in a large margin.

MRC is traditionally defined to be a question-answering task which outputs answers by given inputs of passage-question pairs. Considering the types of answers, Chen (2018) classified MRC's tasks into four categories: cloze-filling of a question with gaps (Ghaeini et al., 2018), multiple-choice from several options (Zhang et al., 2020a), span extraction of answer from the passage (Rajpurkar et al., 2016; 2018; Trischler et al., 2017) and free-style answer generation and summarization from the passage (Nguyen et al., 2016). MRC is regarded to be widely applicable to numerous applications that are rich of question-style queries, such as information retrieval and task-oriented conversations. For detailed survey of this field, please refer to (Zhang et al., 2020c) for recent research roadmap, datasets and future directions.

In this paper, we focus on span-extraction style MRC with unanswerable questions. Rajpurkar et al. (2018) introduced 50K+ unanswerable questions to construct the SQuAD2.0 dataset. Unanswerable questions include rewriting originally answerable questions through ways of negation word inserted or removed, antonym used, entity swap, mutual exclusion, and impossible condition. Plausible answers which correspond to spans in the given passage are attached to these unanswerable questions. Numerous verifiers have been proposed to score the answerability of questions. For example, Hu et al. (2019) proposed a read-then-verify system that explicitly verified the legitimacy of the predicted answer. An answer verifier was designed to decide whether or not the predicted answer is entailable by the input snippets (i.e., segments of the input passage). Their system achieved a F1 score of 74.8% and 74.2% respectively on the SQuAD2.0's dev and test sets. Zhang et al. (2020b) proposed a

---

[1] https://rajpurkar.github.io/SQuAD-explorer/

pipeline with two verifiers: a sketchy reading verifier that briefly investigates the overall interactions of between a segment and a question through a binary classification network following CLM, and an intensive reading module that includes a span extractor and an answerability verifier. These two verifiers are interpolated to yield the final decision of answerablility. This framework has achieved significant improvements (F1 score of 90.9% and 91.4%, respectively on the dev and test sets of SQuAD2.0) than the strong ALBERT baseline (Lan et al., 2020).

Minimizing span losses of start and end positions of answers for answerable questions is overwhelming in current pretraining+fine-tuning frameworks. However, there are still requirements for designing fine-grained verifiers for predicting questions' answerabilities by utilizing the interaction of between answer-injected questions and passages. It is valuable for us to score the linguistic correctness of the predicted answer, through replacing the interrogatives in questions by their predicted answers to check out if it is a linguistically correct sentence and in addition if the given passage can entail that rewritten question. For example, there are two reference answers ("*P is not equal to NP*" as a complete sentence, "*not equal*" as a verb phrase) to a question "*What implication can be derived for P and NP if P and co-NP are established to be unequal?*". When a system predicts "*not equal to NP.*" with unbalanced arguments (i.e., containing objective argument *NP* yet missing subjective argument *P*), it is scored 0 in exact match and discounted in F1 score. Intuitively, "*P is not equal to NP/not equal implication ...*" should be scored higher than "*not equal to NP implication ...*". This motivates our first *position-sensitive question-rewritten verifier*. That is, we score the correctness and completeness of the predicted answer and the rewritten question by the existing CLM models, and build an entailment network that takes cross-attention of between the rewritten question and passage as inputs.

On the other hand, the original passage/document is frequently too long to be directly used in pretraining+tuning frameworks. For example, as reported in (Gong et al., 2020), passages in TriviaQA dataset (Joshi et al., 2017) averagely contain 2,622 tokens generated by BERT tokenizer (Kudo & Richardson, 2018) in their training set. Current CLMs are incapable of accepting arbitrarily long token sequences. Thus, we are forced to cut the passage into segments with fixed length (e.g., 512 tokens with strides such as 128 or 256). Then, the reference input to MRC now is a fix-length segment instead of the whole passage. When the manually annotated answer to the question is out of the scope of the segment, the question will be annotated to be unanswerable regardless its answerability in the whole passage. This brings bias to *answerable* questions since one individual segment *loses its context*, and it is possible that this segment implicitly contain clues for correctly answering the question. For example, in SQuAD2.0, only one answer span is provided for one question regardless of the multiple appearances (104,674 appearances of answer texts for the 86,821 answerable questions) of the same answer text in the given paragraph. For long-text MRC, Gong et al. (2020) proposed recurrent chunking mechanisms by first employing reinforcement learning to centralize the candidate answer span in the segment and then building a recurrent network to transfer contextual information among segments. Considering that it is non-trivial of data-preparation and multi-turn reasoning for training and inferencing under a reinforcement learning framework, we propose a different solution in this paper. We build a hierarchical attention network (HAN) (Yang et al., 2016) on sentence-level, segment-level and finally paragraph-level cross-attentions to questions for extracting and verifying candidate answers. Since the HAN framework includes segment sequences in a recurrent way, Gong et al. (2020)'s recurrent chunking can be regarded as a special case of ours.

We combine these two types of verifiers together into a pipeline, and apply it to three span-extraction MRC benchmark sets: SQuAD2.0 (Rajpurkar et al., 2018), NewsQA (Trischler et al., 2017), and TriviaQA (Joshi et al., 2017) (wikipedia part). Our pipeline achieves significantly better improvements on both exact matching and F1 scores than state-of-the-art baselines.

## 2 ENHANCED LINGUISTIC VERIFIERS

### 2.1 ANSWER-INJECTED REWRITING OF QUESTIONS

The overwhelming types of questions in span-extraction MRC datasets are in a scope of 5W1H, as listed in Table 1. NewsQA's statistical information comes from their website[2]. We calculate the train and validation sets together for SQuAD2.0 and TriviaQA, using these 5W1H interrogatives and their extensions ( such as "whom" and "whose" are all who-style question indicators).

---

[2]https://www.microsoft.com/en-us/research/project/newsqa-dataset

| | what | who | where | how | when | which | other | #total |
|---|---|---|---|---|---|---|---|---|
| SQuAD2.0 | 53.8% | 10.4% | 3.9% | 10.1% | 6.7% | 5.9% | 9.2% | 142,192 |
| NewsQA | 44.0% | 19.2% | 7.1% | 6.8% | 4.1% | 2.2% | 16.5% | 119,633 |
| TriviaQA | 25.4% | 13.5% | 1.5% | 3.2% | 1.3% | 40.8% | 14.3% | 124,876 |

Table 1: Statistical information of question types in the three span-extraction datasets.

There are nearly 5% questions that contain multiple interrogatives, such as $What_{WDT}$ *year and* $where_{WRB}$ *was the first perfect score given?*, $Who_{WP}$ *gets to choose* $where_{WRB}$ *a gymnast starts to run on the runway?*. In order to detect interrogatives, we utilize Stanford dependency parser V4.0.0 (Chen & Manning, 2014) which outputs the dependency tree together with words' POS tags. We inject answer text to questions with one interrogative by simply replacing it with the answer text. For questions with multiple interrogatives, we first parse the answer text and obtain its POS tag sequence and then align the interrogatives with the words in the answer text through simple heuristic rules such as $who_{WP}$ for personal names, $where_{WRB}$ for spaces, $when_{WRB}$ for time and date. In case we fail finding the alignment, we simply attach the answer text at the left-hand-side of the question. We count on multi-head self-attention mechanism (Vaswani et al., 2017) in CLMs as a linguistic scorer to score the linguistic likelihood of the sequence.

## 2.2 REWRITTEN QUESTION RELATED VERIFICATION

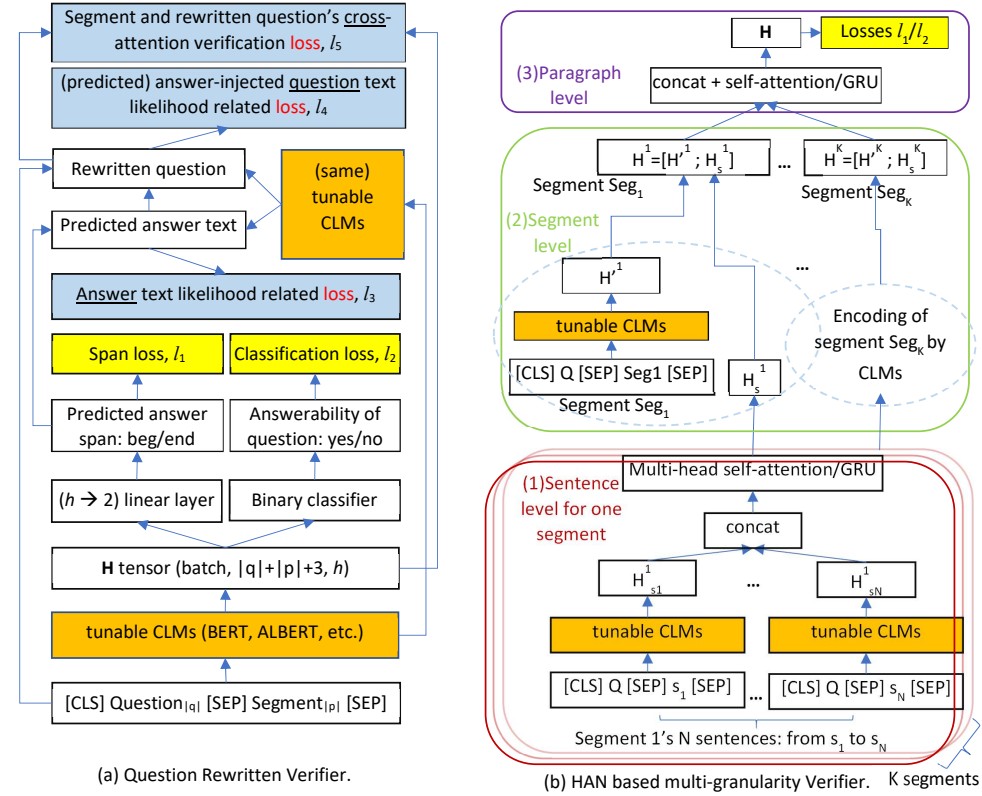

(a) Question Rewritten Verifier.

(b) HAN based multi-granularity Verifier.

Figure 1: Framework of two linguistic verifiers.

Figure 1 illustrates our answer verification framework. We propose two types of maximum likelihood losses to score the sequences of predicted answer text and answer-injected question text by tunable pretrained CLMs. In addition, inspired by (Zhang et al., 2020b), we adapt a cross-attention span loss on the segment and rewritten question.

We first integrate a question (with $|q|$ tokens) and a segment (with $|p|$ tokens) of a paragraph in a standard format of "[CLS] question [SEP] segment [SEP]". Here, [CLS] is a classification token

which is recognized to contain information of the whole sequence with a length of $(|q| + |p| + 3)$. There are two separation [SEP] tokens respectively follow the question and the segment. The whole sequence will be sent to tunable CLMs which first embed the sequence using token, position, and segment (tokens in "[CLS] question [SEP]" are assigned value of 0, otherwise 1) embedding networks. Then, Transformer's encoder layers are employed to represent the sequence to finally output the **H** tensor. Suppose that each token in **H** is represented by a $h$-dimension vector (e.g., $h$=4,096). **H** is sent to a linear layer which projects $h$-dim into 2-dim, i.e., (batch, $|q| + |p| + 3$, 2), standing for the start/end position scores of each token in the segment. Cross entropy loss is adapted to compute the span loss, numbered with $l_1$ in Figure 1. Simultaneously, the vector of the first position in a sequence, i.e., [CLS]'s representation vector is sent to a dropout layer and then a linear layer that projects $h$-dim into 2-dim to score the answerability of the question, by employing cross entropy loss $l_2$.

Our proposed losses are denoted by $l_3$ (phrase-level), $l_4$ (sentence-level) and $l_5$ (question-segment level) in Figure 1. We give detailed ablation experiments of these three losses in the Appendix. Note that we focus on *answerable* questions, i.e., we compute these losses only when the question is answerable annotated in the train/dev sets and predicted to be answerable by the "binary classifier" during training/testing. One motivation is that the best-reported exact matching score for answerable questions only achieved 83.1% while the whole exact matching score was 90.9%, reported in (Zhang et al., 2020b). This 7.8% gap reflects the importance of verifying the linguistic correctness of predicted answer phrases to the answerable questions.

In order to compute the "linguistic likelihood" of these two sequences, predicted answer text and rewritten question, we first attach a [CLS] token to each of them and then sent them to the same CLMs. Each [CLS]'s representative vector will be further sent to a $h$-to-1 linear layer. The second option of computing the likelihood of the predicted answer text is to compare a similarity score (e.g., cosine) of between its [CLS]'s vector and the reference answer text's [CLS] vector.

## 2.3 HAN-BASED VERIFIER

As has been pointed out by Gong et al. (2020), it is problematic to equally cut a paragraph into segments and then separately send (segment, question, answer) triples to the long-text (or, long paragraph/document) MRC system. One drawback is that the appearance of the manually annotated answer in the segment influence the accuracy: best accuracies will be achieved when the answer is in the central of the segment empowered by rich context. Another drawback is that the segments are not communicating with each other during training and their potentially valuable contexts are abandoned unconsciously. In addition, when the answer to the question is out of the scope of the segment, the question will be annotated to be unanswerable in current triple regardless of its paragraph or document -level answerability. Motivated by these, we build a HAN network to make full usage of the linguistic structure of the paragraph, from sentence to segment and finally to paragraph.

We notice that in CLM's forward process, all the tokens in an input sequence "[CLS] Question [SEP] Segment [SEP]" are interacting with each other through multi-head self-attention and feed-forward layers (Vaswani et al., 2017; Devlin et al., 2019; Lan et al., 2020) or a simple bi-directional causal GRU layer[3]. However, we are wondering if this cross-attention of between question and segment still has room to be improved. One motivation is that, masked language modelling of independent sentences and next sentence or sentence permutation prediction all take *sentence* or its pairs as the fundamental linguistic unit during the pretraining processes of CLMs. There are promising results published when applying the sentence selector (Wang et al., 2019) techniques for MRC. Also, during the construction of unanswerable questions in SQuAD2.0 (Rajpurkar et al., 2018), most operations were performed in words of one sentence or several contiguous sentences. These motivate the sentence-layer in our HAN verifier.

We depict our HAN-based multi-granularity verification framework in Figure 1 (b). In order to include current segment-based MRC as a special case of our framework, we still utilize segment here which can be a fixed-length sequence or a natural paragraph in a document. Suppose that there are $N$ sentences in one segment, we first respectively construct sentence-level pairs "[CLS] Q [SEP] $s_i$ [SEP]" and send them to the tunable CLMs, then we concatenate the yielded tensors $\mathbf{H}^1_{s_i}$ and introduce a multi-head self-attention or GRU layer to perform cross-sentence interactions.

---

[3]We report results on bi-GRU due to its better accuracies and smaller size of parameters compared with self-attention layers. The extra cost is an averagely +30% of train/test time.

Through this way, we obtain a tensor $\mathbf{H}_s^1$. Q here represents the original question. Note that, during concatenating, vectors of [CLS] and two [SEP] tokens in different pairs will be added up into three new vectors, tokens in Q(uestion) will be added up following their positions and tensors of $s_1$ to $s_N$ are concatenated to form a tensor expressing the same number of tokens (e.g., denoted as $|p_1|$) of their original segment. Thus, the result shape after concatenating is (batch, $|q| + |p_1| + 3, h$).

At the same time, we make use of the whole segment S to form a sequence mentioned formerly, "[CLS] Q [SEP] S [SEP]" and it will be sent to the tunable CLM to obtain another tensor $\mathbf{H}'^1$. These two tensors, $\mathbf{H}'^1$ and $\mathbf{H}_s^1$, are added up together (by keeping the same number of tokens) to form a final representation for current segment's interaction with the question. A second concatenation followed by a cross-segment level multi-head self-attention or GRU layer will be used again to obtain paragraph/document level representation tensor, $\mathbf{H}$, with a shape of (batch, $|q| + |p| + 3, h$). The consequent steps are quite alike computing span/answerability losses $l_1$ and $l_2$.

In particular, this framework will be retrograded to be identical to the frequently used pretraining+fine-tuning methods (Devlin et al., 2019; Lan et al., 2020; Gong et al., 2020), when we remove the sentence and paragraph levels. One benefit in this framework is that, we no longer need to judge a question to be unanswerable in case the answer span is not fallen in current segment's region. We take the whole document into consideration now. Another benefit of this framework is that, we bypass a direct modelling (with obstacles of GPU memory constraints and computing complexities) on the long document through sentence, segment and paragraph cross-attentions with the question.

## 2.4 VERIFIER COMBINATION

Our two verifiers can both perform span extraction and answerability prediction. We thus train these two verifiers independently and combine them together by following (El-Geish, 2020).

## 3 EXPERIMENTS

We perform experiments independently and jointly on these two verifiers. We select three span-extraction MRC benchmark sets: SQuAD2.0 (Rajpurkar et al., 2018), NewsQA (Trischler et al., 2017), and TriviaQA (Joshi et al., 2017) (wikipedia part). We implement our verifiers under Huggingface's transformers[4] written in PyTorch. We specially selected ALBERT (Lan et al., 2020) pretrained model of "albert-xxlarge-v2"[5] as our tunable CLM. Loss functions are optimized by the Adam algorithm with weight decay (Kingma & Ba, 2015; Loshchilov & Hutter, 2019). ALBERT's tokenizers are reused to tokenize words into word pieces (Kudo & Richardson, 2018). We run experiments on three machines, each with a NVIDIA V100 GPU card with 32GB memories.

We respectively set the maximum sequence, question, answer and segment lengths to be 512, 64, 30, and 445. The document stride is set to be 128. The batch size for training is set to be 1 with a gradient accumulation step to be 24. We set the learning rate to be 3e-05 with 814 warm-up steps. Weight decay is set to be 0. We separately train 4 epochs on each dataset. For SQuAD2.0 and TriviaQA, we report results on the development set, and for NewsQA, we report both development and test sets' result since its test set is open.

## 3.1 BENCHMARK DATASETS

**SQuAD2.0** As a famous and widely used MRC benchmark dataset with unanswerable questions, SQuAD2.0 (Rajpurkar et al., 2018) updates the 100K questions in their earlier version of SQuAD1.1 (Rajpurkar et al., 2016) with over 50K new, unanswerable questions that are written adversarially by crowdworkers by referring answerable ones. As former mentioned, most of the unanswerable questions are distortion and rewriting of some specific words by referring limited sentences in the paragraph. The training set contains 87K answerable questions with one answer annotated and 43K unanswerable questions with plausible answers provided.

**NewsQA** (Trischler et al., 2017) is a question-answering dataset on paragraphs of news articles which are longer than that in SQuAD2.0, averagely 600 words. The training dataset has 20K

---

[4]https://github.com/huggingface/transformers
[5]https://github.com/google-research/ALBERT

| Model | Dev | | Test | |
|---|---|---|---|---|
| | **EM** | **F1** | **EM** | **F1** |
| Regular Track (i.e., results reported in papers) | | | | |
| Joint SAN (Liu et al., 2018) | 69.3 | 72.2 | 68.7 | 71.4 |
| U-Net (Sun et al., 2018) | 70.3 | 74.0 | 69.2 | 72.6 |
| RMR+ELmo+Verifier (Hu et al., 2019) | 72.3 | 74.8 | 71.7 | 74.2 |
| Retro-Reader (Zhang et al., 2020b) | 87.8 | 90.9 | 88.1 | 91.4 |
| Top results on the leaderboard (`https://rajpurkar.github.io/SQuAD-explorer/`) | | | | |
| Human | - | - | 86.8 | 89.5 |
| XLNet (Yang et al., 2019) | 86.1 | 88.8 | 86.4 | 89.1 |
| RoBERTa (Liu et al., 2019) | 86.5 | 89.4 | 86.8 | 89.8 |
| ALBERT (Lan et al., 2020) | 87.4 | 90.2 | 88.1 | 90.9 |
| ALBERT+ Entailment DA Verifier (single model), CloudWalk | - | - | 87.8 | 91.3 |
| ALBERT+verifier, aanet_v2.0 (single model), QIANXIN | - | - | 88.4 | 91.0 |
| SA-NET on Albert (ensemble), QIANXIN | - | - | **90.7** | **93.0** |
| Retro-Reader online (Zhang et al., 2020b) | - | - | 90.6 | **93.0** |
| Question-rewritten verifier (single, ours) | **89.1** | **91.9** | 89.7 | 91.6 |
| HAN verifier (single, ours) | **89.6** | **92.5** | 90.2 | 92.2 |
| Combination (ensemble, ours) | **90.5** | **93.3** | **90.7** | **93.0** |

Table 2: The results (%) from single/ensemble models for the SQuAD2.0 challenge.

| Methods | HasAns | | NoAns | | All | |
|---|---|---|---|---|---|---|
| | **EM** | **F1** | **EM** | **F1** | **EM** | **F1** |
| Retro-Reader (Zhang et al., 2020b) | 83.1 | 89.4 | 92.4 | 92.4 | 87.8 | 90.9 |
| Question-rewritten | 86.9 | 92.5 | 91.3 | 91.3 | 89.1 | 91.9 |
| Question-rewritten-$l'_3$ | 86.7 | 92.3 | 91.3 | 91.3 | 89.0 | 91.8 |
| HAN | 86.9 | 92.7 | 92.3 | 92.3 | 89.6 | 92.5 |
| HAN without sentences | 86.4 | 91.8 | 92.0 | 92.0 | 89.2 | 91.9 |
| Combination (ensemble) | 87.6 | 93.2 | 93.4 | 93.4 | 90.5 | 93.3 |

Table 3: Detailed results (%) of our two types of verifiers on the SQuAD2.0 dev set.

unanswerable questions among 97K questions. Generally, current results on NewsQA are much lower than SQuAD2.0, in a range of 60% for exact match and 70% for F1 scores.

**TriviaQA** (Joshi et al., 2017). TriviaQA is an MRC dataset containing over 650K question-answer-evidence triples. TriviaQA includes 95K question-answer pairs authored by trivia enthusiasts and independently gathered evidence documents, 6 per question on average, that provide high quality distant supervision for answering the questions. We specially use the wikipedia part of this data.

### 3.2 EVALUATION METRICS AND SIGNIFICANT TESTS

Following most of the former researches, we use two official metrics to evaluate the performances of our verifiers: (1) Exact Match (**EM**) of scoring 1 for predicted answer text exactly matching with its reference and 0 otherwise, and (2) a softer metric **F1** score that measures the weighted average of the precision and recall rates at word level regardless of punctuation. We perform the significant test following Zhang et al. (2020b), by taking EM as the goodness measure under McNemar's test (McNemar, 1947), which was designed for paired nominal observations, and is appropriate for binary (1/0) classification tasks (Ziser & Reichart, 2017).

### 3.3 RESULTS ON SQUAD2.0

In Table 2, the results except ours are obtained by the online codalab evaluation server and the corresponding literatures (if have). The best results are in bold font. Our models are significantly better than all the baselines ($p < 0.05$ for the individual verifiers and $p < 0.01$ for the combination) with published papers on the development set. In the test set, our best submission achieved the same EM and F1 scores with the top-1 in the leaderboard. We further check the accuracies on the answerable subset since one of our motivations is to explicitly verifier the textual answers. In addition, we perform ablation tests on several optional conditions. The results are listed in Table 3.

| Model | Dev | | Test | |
|---|---|---|---|---|
| | **EM** | **F1** | **EM** | **F1** |
| BARB (Trischler et al., 2017) | 36.1 | 49.6 | 34.1 | 48.2 |
| Match-LSTM (Wang & Jiang, 2017) | 34.4 | 49.6 | 34.9 | 50.0 |
| BiDAF (Seo et al., 2017) | - | - | 37.1 | 52.3 |
| FastQA (Weissenborn et al., 2017) | 43.7 | 56.4 | 41.9 | 55.7 |
| FastQAExt (Weissenborn et al., 2017) | 43.7 | 56.1 | 42.8 | 56.1 |
| AMANDA (Kundu & Ng, 2018) | 48.4 | 63.3 | 48.4 | 63.7 |
| DECAPROP (Tay et al., 2018) | 52.5 | 65.7 | 53.1 | 66.3 |
| BERT (Devlin et al., 2019) | - | - | 46.5 | 56.7 |
| NeurQuRI (Back et al., 2020) | - | - | 48.2 | 59.5 |
| Retro-Reader (Zhang et al., 2020b) | 58.5 | 68.6 | 55.9 | 66.8 |
| Question-rewritten verifier (single, ours) | **60.5** | **70.6** | **58.3** | **68.1** |
| HAN verifier (single, ours) | **62.1** | **71.5** | **60.2** | **69.5** |
| Combination (ensemble, ours) | **63.2** | **72.9** | **61.8** | **70.7** |

Table 4: The results (%) from single/ensemble models for the NewsQA dataset.

| Methods | HasAns | | NoAns | | All | |
|---|---|---|---|---|---|---|
| | **EM** | **F1** | **EM** | **F1** | **EM** | **F1** |
| Question-rewritten (single) | 60.1 | 74.1 | 61.4 | 61.4 | 60.5 | 70.6 |
| HAN (single) | 60.9 | 73.8 | 65.4 | 65.4 | 62.1 | 71.5 |
| Combination (ensemble) | 61.1 | 74.3 | 69.5 | 69.5 | 64.1 | 72.9 |

Table 5: Our detailed results (%) of the two types of verifiers on the NewsQA dev set.

For the detailed comparison, the best baseline with open results is Retro-Reader tuned on ALBERT (Zhang et al., 2020b). We thus list their results here for a direct comparison. We separately run two variants for our first question-rewritten verifier by removing $l'_3$ from the loss function of $l$ (Equation 7). The results slightly dropped yet not significant, reflecting a less importance of directly computing cosine similarities of between the predicted answer and the rewritten question. We in addition remove the sentence-level in our HAN verifier, the results significantly dropped ($p < 0.05$), especially for the answerable questions' accuracies. This reflects the effectiveness of the sentence level cross-attention of between the sentences in segments and the questions. Compared with the results of Retro-Reader, our system performs significantly better in terms of verification of *answerable* questions, such as the EM score is improved 4.5% of from 83.1% to 87.6%. These improvements align with our initial motivation of designing verifiers for answerable questions.

## 3.4 RESULTS ON NEWSQA

In Table 4, our model results are in bold font, which is significantly better than all the baselines ($p < 0.01$ for the individual verifiers and their combination) on the development and test sets.

We further check the accuracies on the answerable subset since one of our motivations is to explicitly verifier the textual answers. In addition, we perform ablation tests on several optional conditions. The results are listed in Table 5. Most observations from Table 5 align with the results of SQuAD2.0. The results for answerable questions are stably increasing of the three variants. The results for unanswerable questions are rather less-stable: HAN's result is the best and when the two verifiers are combined together, we lose some benefits.

## 3.5 RESULTS ON TRIVIAQA

It is difficult to directly compare our results with systems under the TriviaQA dataset, due to limited published papers in their official competitions. We instead align with the recent published paper (Gong et al., 2020) which is aiming at solving quite similar problems of cross-segmentation. In Table 6, our model results are in bold font, which is significantly better than three list baselines ($p < 0.01$ for both the individual verifiers and their combination) on the development set.

Similarly, we further check the accuracies on the answerable subset since one of our motivations is to explicitly verifier the textual answers. In addition, we perform ablation tests on several optional

| Models | HasAns | | NoAns | | All | |
|---|---|---|---|---|---|---|
| | EM | F1 | EM | F1 | EM | F1 |
| BERT-Large (Devlin et al., 2019) | 56.4 | 62.6 | 58.3 | 58.3 | 57.0 | 61.3 |
| Sentence-Selector (Wang et al., 2019) | 54.5 | 60.6 | 58.1 | 58.1 | 55.6 | 59.8 |
| BERT-RCM (Gong et al., 2020) | 56.7 | 62.8 | 62.9 | 62.9 | 58.6 | 62.9 |
| ALBERT (Fine-tuned) (Lan et al., 2020) | 57.3 | 65.4 | 63.4 | 63.4 | 59.2 | 64.8 |
| Retro-Reader + ALBERT (Fine-tuned) (Zhang et al., 2020b) | 58.2 | 64.7 | 65.1 | 65.1 | 60.3 | 64.8 |
| RAG (Fine-tuned, Open-Domain) (Lewis et al., 2020) | - | - | - | - | - | **68.0** |
| T5-11B+SSM (Fine-tuned, Closed-Book) (Raffel et al., 2019) | - | - | - | - | - | 60.5 |
| GPT-3 Few-Shot (Brown et al., 2020) | - | - | - | - | - | **71.2** |
| Question-rewritten verifier (single, ours) | 59.9 | 66.3 | 59.2 | 59.2 | **59.7** | 64.1 |
| HAN verifier (single, ours) | 60.2 | 66.6 | 61.6 | 61.6 | **60.6** | 65.0 |
| Combination (ensemble, ours) | 59.5 | 65.9 | 65.6 | 65.6 | **61.4** | 65.8 |

Table 6: The results (%) for the TriviaQA dataset. ALBERT=ALBERT-xxlarge-v2.

conditions. Most observations from Table 6 align with the results of SQuAD2.0 and NewsQA. The results for answerable questions are stably increasing of the three variants. These further show that our cross-segment proposal under HAN is effective for verifying answers for answerable questions. Finally, three stronger baselines are listed for reference: Retrieval-Augmented Generation (RAG) (Lewis et al., 2020) which takes the whole wikipedia as a reference, T5 (Raffel et al., 2019) with richer pre-training datasets and GPT-3 (Brown et al., 2020) on 570GB datasets and 175 billion parameters.

## 4 RELATED WORK

Due to space limitation, we are not able to review all the previous work as listed in Table 2, 4 and 6. We only select the state-of-the-art baselines (with open reference papers) for a brief comparison.

Since the releasing of SQuAD2.0 data with unanswerable questions, there are a number of research on proposing novel verifiers focusing on the prediction of unanswerable questions, such as the read+verify framework (Hu et al., 2019) and the retro-reader framework (Zhang et al., 2020b). Our research fills in the group of constructing verifiers for span-extraction MRC. Instead of concentrating on building verifiers for unanswerable questions, we propose linguistic verifiers on the predicted answer texts and the answer-injected questions. The target is to further leverage linguistic information to constraint the answer texts to better fit the given question and the paragraph.

On the other hand, little research has been performed on cross-segmentation verification for long-text MRC. Compared with recent work (Gong et al., 2020), we propose a HAN based model that employs sentence-level, segment-level and finally paragraph-level cross-attention information with the questions. This proposal both bypasses the strong bias assumption of assigning questions to be unanswerable when current segment does not contain the answer span (instead of the answer text) and introduces richer linguistically structural information to the whole framework. Experiment results on three datasets show that our verifiers can significantly exceed strong state-of-the-art baselines.

## 5 CONCLUSION

In this paper, we propose two types of verifiers for *answerable* questions in span-extraction MRC. Our contributions include, (1) we propose a rewritten question oriented verifier that aims at improving the linguistic correctness of the predicted answers conditioned on its question and the given paragraph; (2) we propose a HAN verifier to deal with the long-paragraph problem to transfer contextual information across segments and to bypass the obstacle of hard decisions to the answerability of a question when current segment does not contain the annotated answer; (3) we report significant improvements on three widely used benchmark sets comparing our verifiers with state-of-the-art baselines. In the future, we are planning to enrich our verifiers for open-domain MRC where long-text documents are frequent and it will be harder to exactly define "unanswerable" questions and deep representative information retrieval techniques should be taken into consideration.

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

## A    Appendix

### A.1    Network of the Question-Rewritten Verifier

Formally, the whole framework depicted in Figure 1 is expressed by the following equations:

$$\mathbf{H} = \text{CLM}([\text{CLS}] \ Q \ [\text{SEP}] \ S \ [\text{SEP}]),$$

$$\mathbf{H}_1 = \text{Linear}^1_{h \to 2}(\mathbf{H}),$$

$$l_1 = \text{Loss}(\mathbf{H}_1, \text{ref.span}), \tag{1}$$

$$\mathbf{H}_2 = \text{Linear}^2_{h \to 2}(\text{dropout}(\mathbf{H}[:, 0])),$$

$$l_2 = \text{Loss}(\mathbf{H}_2, \text{ref.answerability}). \tag{2}$$

Here, Q and S respectively stand for the question and the segment, ref is reference, ans is (predicted) answer. $l_1$ and $l_2$ are cross entropy losses of span correctness and answerability.

The likelihood losses are computed by:

$$\text{span} = \text{argmax}(\mathbf{H}_1),$$

$$\text{ans.text} = S[\text{span}],$$

$$\mathbf{H}_3 = \text{CLM} \ ([\text{CLS}] \ \text{ans.text}),$$

$$m = \text{Linear}^3_{h \to 1}(\text{dropout}(\mathbf{H}_3[:, 0])),$$

$$l_3 = -\log(m). \tag{3}$$

Alternatively, we can directly compute the similarity between the predicted answer and its reference:

$$\mathbf{H}'_3 = \text{CLM} \ ([\text{CLS}] \ \text{ref.text}),$$

$$m_{ref} = \text{Linear}^3_{h \to 1}(\text{dropout}(\mathbf{H}'_3[:, 0])),$$

$$l'_3 = -\text{cosine}(m, m_{ref}). \tag{4}$$

The linguistic likelihood of the rewritten question is computed by:

$$Q' = \text{Injection}(Q, \text{ans.text}),$$

$$\mathbf{H}_4 = \text{CLM}([\text{CLS}] \ Q'),$$

$$n = \text{Linear}^4_{h \to 1}(\text{dropout}(\mathbf{H}_4[:, 0])),$$

$$l_4 = -\log(n). \tag{5}$$

Here, scores $m$ and $n$ are likelihood of the predicted answer text and rewritten question text computed by CLM. Operation [:,0] is to ranger over sequences in a batch to pack the [CLS] token's representation vector.

We finally formulate the cross-attention of between the segment and rewritten question:

$$\mathbf{H}' = \mathbf{H} + \text{RELU}(\text{Softmax}(\mathbf{H}\mathbf{H}_4^\top)\mathbf{H}_4),$$

$$\mathbf{H}'_1 = \text{Linear}^5_{h \to 2}(\mathbf{H}'),$$

$$l_5 = \text{Loss}(\mathbf{H}'_1, \text{ref.span}), \tag{6}$$

$$l = l_1 + l_2 + l_3 + l'_3 + l_4 + l_5. \tag{7}$$

We apply a residual adding when computing $\mathbf{H}'$. The span loss is computed again on this $\mathbf{H}'$ after a linear projection. Losses from $l_1$ to $l_5$ will be added up together as their final loss $l$.

### A.2    Network of the HAN Verifier

A formal description of this framework can be concluded as:

$$\mathbf{H}^k_{s_i} = \text{CLM}([\text{CLS}] \ Q \ [\text{SEP}] \ s_i \ [\text{SEP}]),$$

$$\mathbf{H}^k_s = \text{SelfAtten}(\text{concat}(\mathbf{H}^k_{s_1}, ..., \mathbf{H}^k_{s_N})),$$

$$\mathbf{H}'^k = \text{CLM}([\text{CLS}] \ Q \ [\text{SEP}] \ S^k \ [\text{SEP}]),$$

$$\mathbf{H}^k = \mathbf{H}'^k + \mathbf{H}^k_s.$$

| Methods | HasAns | | NoAns | | All | |
|---|---|---|---|---|---|---|
| | EM | F1 | EM | F1 | EM | F1 |
| Retro-Reader (Zhang et al., 2020b) | 83.1 | 89.4 | 92.4 | 92.4 | 87.8 | 90.9 |
| Question rewritten | 86.9 | 92.5 | 91.3 | 91.3 | 89.1 | 91.9 |
| Question rewritten - $l'_3$ | 86.7 | 92.3 | 91.3 | 91.3 | 89.0 | 91.8 |
| Question rewritten - $l_3$ | 86.2 | 91.6 | 91.2 | 91.2 | 88.7 | 91.4 |
| Question rewritten - $l_4$ | 85.2 | 90.6 | 91.0 | 91.0 | 88.1 | 90.8 |
| Question rewritten - $l_5$ | 84.6 | 90.0 | 90.8 | 90.8 | 87.7 | 90.4 |

Table 7: Ablation results (%) of the question rewritten verifier's $l_3/l_4/l_5$ on the SQuAD2.0 dev set.

These equations describe sentence-level and segment-level operations for a segment $k$, with $N$ sentences. For the paragraph-level with $K$ segments:

$$\mathbf{H} = \text{SelfAtten}(\text{concat}(\mathbf{H}^1, ..., \mathbf{H}^K)).$$

We can reuse the operations for computing $l_1$ and $l_2$ by taking $\mathbf{H}$ as input.

### A.3 ABLATION STUDY OF $l_3$, $l_4$ AND $l_5$

We proposed three types of loss functions in our question rewritten verifier (Figure 1), $l_3$ for evaluating the correctness and completeness of a predicted answer phrase, $l_4$ for scoring the linguistic correctness of a rewritten question, and $l_5$ for jointly "similarity" computing of between the rewritten question and the segment. Easy to observe that these three loss functions are designed in a small-to-big granularity of number of tokens involved.

We performed an ablation study of the impacts of these losses to the final results. We select SQuAD2.0 as our test-bed and report results on its development set, as listed in Table 7.

After removing individual losses, the EM/F1 scores dropped significantly ($p$ ¡ 0.05 testing on EM). Specially, $l_5$ plays the most important role of evaluating the relationship of between rewritten question and a segment. $l_4$ contributes more than $l_3$ due to the fact that $l_4$ is used to reflect the general correctness of the rewritten question while $l_3$ is only for the individual answer phrase with rather limited information of "linguistic context".

