# OpenReview forum: "Machine Reading Comprehension with Enhanced Linguistic Verifiers"
_ICLR.cc/2021/Conference — Reject_

### Official Review · AnonReviewer1 · 2020-10-26
**Official Blind Review #1**

**Rating:** 6
**Confidence:** 5

**Review:**

This work addresses two main challenges of span-extraction style machine reading comprehension (MRC) tasks: how to evaluate the syntactic completeness of predicted answers and how to utilize the rich context of long documents. To handle such challenges, Question Rewritten Verifier (QRV) and Hierarchical Attention Network (HAN) are proposed respectively. The former uses a question-rewritten method to replace the interrogatives in the question with the answer span for further verification. The latter adopts a hierarchical multi-granularity (sentence, segment, and paragraph) cross-attention to extract and utilize the context information of long paragraphs. Compared with the strong baselines, both the verifiers and their combination achieved relatively significant accuracy improvement on three mainstream span-extraction MRC tasks: SQuAD2.0, NewsQA, and TriviaQA.

-------------------------------------------
Strengths:

1. The idea of bringing the answer back to the question for further validation is sound and it is reasonable for humans to do this process to verify the candidate answer in real-world practice.

2. The question rewritten strategy is simple and effective, which brings improvements. HAN also handles the problem of long sequence well.

3. The overall method achieves state-of-the-art results. The significance test shows significant improvements over baselines.

-------------------------------------------
Weaknesses:

1. The design of the training target (loss) in QRV is complex and not interpretable enough. There are many loss functions. How about their contributions to the final performance?

2. There is no test result reported for SQuAD2.0, though it is possible to obtain the results without making it public. Therefore, the clarity, “Due to anonymous issues, we have not submitted our results in an anonymous way to obtain results on the hidden test set.”, is not quite convincing.

3. The improvement of accuracy is mainly reflected in the questions of HasAns, which has no obvious contribution to the recognition accuracy of NoAns, which is one of the main challenges of the current MRC tasks.

-------------------------------------------
Questions:

1. (Section 1 page 2 line 26) The paragraphs are divided into segments, with fixed length (e.g.,512 tokens with strides such as 128 or 256) and then divides the segment into sentences. So when dividing the paragraph, what if the dividing point is in the middle of a sentence? Would the incomplete sentence be discard？If not, how to further divide the segment to sentence level? Further clarification of the process would be beneficial.

2. (Section 2.1 page 3 line 8) When failing to find the alignment, the answer text is attached at the left-hand side of the question. It obviously damages the sentence structure. So will this affect the judgment of the model in the following process? In another word, would it have an impact on the performance of the final model (Increase or decrease) if question-written including subsequent loss calculations were not done on such questions?

3. (Section 2.2 page 4 line 17) Multiple losses are employed, but the paper did not distinguish the practical effectiveness of each loss. My concern is whether each of the objectives is necessary since the experiment results in Table 3 has verified that  $l’_{3}$ does not significantly improve the accuracy of the model. Would the authors further verify the contribution of other losses to model performance (except for apparently indispensable l1 and l2)?

4. (Figure 1 (b)) Do the tunable CLMs of sentence-level and segment level share parameters? Besides, may the authors list the number of parameters of each model (QRV, HAN, and Combination)?

----------------------------------
Minor Issues:

The citation format is not consistent, please check the usage of \citep{} and \citet{}.

---

> ### Author Response · Authors · 2020-11-17
> **Thank you so much for the precious and constructive questions and comments**
>
> 1. Sorry for the ambiguity. L3 (loss 3, we will use L1-5 to express losses 1 to 5) to L5 are from small to big: L3 (phrase-level) for the answer phrase LM score, L4 (sentence-level) for the rewritten question LM score and L5 (multi-sentence-level) for the cross-attention loss of between a segment and the rewritten question. Our current ablation tests show that their contribution to the final performance roughly aligns with their granularity: L3<L4<L5. For sure, detailed scores will be included in a future submission.
> 2. We submitted our models un-openly to SQuAD2.0 and the best model achieved EM/F1 (%) of 88.631/93.245 of answerable questions, 92.821/92.821 of unanswerable questions and finally 90.679/93.038 of the whole test set. The EM score is slightly worse than the top-1 result of 90.724 in the Leaderboard (2020/Nov/17), yet the F1 score is slightly better than the top-1 result of 93.011. [These results are just for reference. We totally understand the policy of reviewing and these results may not necessary be taken into consideration for paper evaluation].The code is under cleaning and we are planning to release the code in a near future.
> 3. Yes, we initially aimed at verifying the syntactic correctness of answers for answerable questions. Our initial motivation was to help solving the open-domain QA applications: when a search engine received a question, first applying information retrieval methods to constraint the candidate passages and then use machine reading modules to further detect the exact phrase as the answer. When we have this first IR and then MRC pipeline of open-domain MRC, we would like to appeal that: (1) supplying complete and linguistically correct answer phrases by retrieving and reasoning from a list of candidate passages/documents is as important as (2) judging a question to be unanswerable under a given passage. This is because, we will have quite many candidate passages for comparison: a question not answerable in one passage do not necessarily mean it can not be answerable in another passage of different document. We also notice that MRC models are applied to Google/Bing for deep question-answering. Furthermore, based on our query log analysis: (1) the amount of answerable questions is significantly larger than that of unanswerable questions and (2) disambiguating plausible answers is more challenge than judging a question to be unanswerable. Intuitively speaking, we are intended to obtain answers or unknown knowledge when we submit a question-style query to IR. In addition, we possibly need a reason when a question is judged to be unanswerable. So, hopefully, the explainable classification of unanswerable questions will be a helpful direction.
> Finally, even we did not design it intentionally, in Table 3, the NoAns’s EM/F1 are still comparable (92.6 vs. 92.4) to that of Retro-Reader which is specially designed for NoAns.
>
> Answer to Q1:  As far as we know, most baselines simply segment by fixed token length regardless of the sentence completeness. In our code, when the segmentation position is in the middle of a sentence, we choose to put that full sentence into the next segment. Thus, each segment in our input only contains complete sentences.
>
> Answer to Q2: Appreciate for this insightful question. When we simply attach the predicted answer text to the question, it does break the independent sentence structure of the original question and yields a worse loss of L4. On the other hand, our cross-attention loss L5, can help alleviating this, which is designed to give a guidance of the dependency of between the words in the rewritten question and the segment. We investigated this and find the results were almost the same, also considering that only 2% questions in total have this alignment issue.
>
> Answer to Q3: Appreciate! Please also refer to our first answer, and yes we do need to append the detailed ablation results.
>
> Answer to Q4: Yes, we employed the same ALBERT-xxlarge as the unique tunable CLM for both sentence-level and segment-level representative learning.
> In QRV, only O(1) parameters for L3 and L4. For L5, we used a revised one-layer multi-head attention layer same to (Zhang et al., 2020b)  with 4 Linear networks and 4*(512 * 512+512)=1,050,624 parameters, where 512 * 512 for weight w and 512 for bias b. In HAN, we additionally have two GRU RNN for sentence-level and segment-level recurrent sequence learning, each with max seq_len=64 and hidden vector dim=512, so it is ((512+64)*64+64+64) * 3 * 2=221,952 parameters. In addition, we tried one-layer self-attention in stead of GRU, yet the results were significantly worse than that of using GRU. Thus, we reported results on GRU instead of one-layer self-attention. Multi-layer self-attentions are to be testified in the future. In the "combination", we additionally have 536,976 parameters (https://arxiv.org/pdf/2004.07067.pdf Figure 3).
>
> In addition, we will recheck \citep and \citet, appreciate for your time.

---

### Official Review · AnonReviewer4 · 2020-10-29
**Good results, demonstrates the value of explicit verification/hierarchies in DL, lower novelty wrt ML elements, manuscript needs significant revision**

**Rating:** 5
**Confidence:** 3

**Review:**

MACHINE READING COMPREHENSION WITH ENHANCED LINGUISTIC VERIFIERS

The authors propose two linguistic verifiers for improving extractive question answering performance when the question is answerable. The first replaces interrogatives in the question (who etc.) with candidate answers and evaluates this both in isolation and in combination with the answer-containing sentence to do answer verification. The second verifier jointly encodes individual sentences and spans with questions in a hierarchical manner to improve answer prediction performance. Solid gains on Squad, NewsQA, and TriviaQA are reported for both methods when applied in isolation, and in combination.

Strengths:

- The techniques are sound and lead to solid gains on 3 benchmark datasets.
- The approaches, while relatively straightforward, illustrate that explicit verification and hierarchical evaluation continue to improve application results, despite the high capacity and efficacy of the SOTA deep architectures.

Limitations:

- The paper is understandable but the presentation could be significantly improved. Figure 1a for example, is a bit overwelming, and should probably be replaced with something more focused, and moved into supplementary material. Several sentences I couldn't understand, for example "Minimizing span losses of start and end positions of answers for answerable questions is overwhelming in current pretraining+fine-tuning frameworks." Overall I feel that the paper could use some additional polishing.
- A similar hierarchical (HAN) approach was previously proposed for verifying unanswerable questions, but their approach for answerable questions appears to be more effective.
- The paper has lower novelty wrt ML elements. The component architectures/models that make up their system are well established.
- The replacing of interrogatives with the answer and the associated rules for doing so feel like they have somewhat limited scope (e.g. factoid questions, single interrogative questions, etc.). When there is more than one interrogative, the authors back off to simply appending the answer to the question... perhaps this can be done all the time without compromising the performance gains?
- Verification (esp. for the HAN verifier, where extra forward passes are done for each sentence and sub-paragraph) is more more computationally demanding, but this is not discussed.

Overall Assessment:

A solid applications paper on extractive question answering. However, I feel that the paper is perhaps better suited for an NLP-application focused audience (e.g. NAACL, deadline approaching), since the results are strong, but the paper has lower novelty wrt core ML. Furthermore, the manuscript is in need of significant revision before it can be considered for acceptance at ICLR.

quality 5/10 (+results on multiple benchmark datasets, -manuscript needs substantial revision)
clarity 5/10 (+understandable for the most part, -manuscript/figures not clear in many places)
originality 6/10 (+novel approaches to QA verification, -lower novelty wrt ML elements)
significance 6/10 (+strong QA results, +demonstrates value in explicit verification/hierarchical processing in DL applications, -perhaps more suitable for an NLP-applications focused audience)
overall (5)

Post-rebuttal:

Authors, thank you for your feedback. The additional results around relative speed and performance have strengthened the paper. However, I still feel that the paper still needs significant polishing before final publication (figures, grammar, presentation), and that the paper is better suited for an NLP-focused conference, and so I have not updated my final score.

---

> ### Author Response · Authors · 2020-11-17
> **We appreciate your time for reviewing this paper and your detailed comments and questions**
>
> 1. Please allow us to give a brief introduction of the MRC task. For this sentence “minimizing span losses”, are “span loss, 1” in Figure 1a, i.e., the start/end positions in a segment (or, paragraph). In current MRC models, one major task is to find the answer text from the segment by given a pair of <question, segment>. Every token in the segment has an index position such as from 0 to 511. In addition, a reference answer in the input is expressed by a pair of start/end index positions, such as [5,8] stands for the 5-th to 8-th words are the answer phrase. We can consequently design binary classification losses by comparing the predicted start/end positions with the reference start/end positions.
>
> 2. We appreciate if we can be shared with the reference and make a full comparison. Currently, we used HAN and MRC to search in Google, and found some related papers: (1) https://web.stanford.edu/class/archive/cs/cs224n/cs224n.1194/posters/15791528.pdf, (2) https://www.aclweb.org/anthology/P18-1158.pdf (3) https://www.aaai.org/ocs/index.php/AAAI/AAAI18/paper/viewFile/16331/16177 and (4) https://stefanheinrich.net/files/2019_Alpay_IJCNN.pdf. Not sure if these references are related or not. We had a learn of these four papers and even all of them mentioned HAN and related variants, for different MRC tasks (of multi-choice) or based on different CLMs, with much worse results then our baselines.
>
> 3. We initially aimed at NLP applications of learning representation algorithms/models. Still, we cautiously appeal our work for the impact of a challenging NLP domain of machine reading comprehension, especially for open domains. MRC is adapted in modern search engines such as Google/Bing/Baidu and MRC is testified to be beneficial to a list of NLP tasks such as NER, text generation, conversations, QA, etc.
>
> 4. Thanks for the insightful question. The question is also related to the ablation test of the effectiveness of loss 4 which is using CLMs to evaluate the rewritten question. [please also refer to our comment 1 and “Answer to Q2” to “AnonReviewer1” ] We compared the suggested left-hand-side append without replacing the interrogatives and found that the result dropped significantly due to several facts: (1) most answers are phrases with multiple words and their boundaries are more difficult to be scored if we simply attach them to the head of the question: the context of the question will be less helpful to detect missing or duplicating of words in the answer phrase; (2) answer phrases make it more difficult for the cross-attention loss to find dependency relations of between the interrogative words and the answer phrases. Also note that only around 2% questions were failed of alignment and attached the predicted answers in the left-hand-side.
>
> 5. Please also refer to our “answer 3 for AnonReviewer 2” for time complexity reports. Also, for the increasing of parameters, please refer to our “Answer to Q4 to AnonReviewer1”. Due to the recurrent networks of bidirectional GRUs are causal and difficult to be computed in parallel, the time complexity for both training and testing was increased largely. In addition, we tried one-layer self-attention in stead of GRU, yet the results were significantly worse than that of using GRU. Thus, we reported results on GRU instead of one-layer self-attention. In the future, it will be reasonable to consider transformer’s multi-layer sequence modeling methods or their variants to further decrease the time complexity.

---

### Official Review · AnonReviewer2 · 2020-10-30
**Official Blind Review #2**

**Rating:** 5
**Confidence:** 4

**Review:**

This paper proposes two types of linguistic verifiers for machine reading comprehension task in span extraction form. One is a rewritten question oriented verifier that checks the linguistic correctness of the extracted answers, and the other is based on a hierarchical attention network for answerability classification and boundary determination. The two verifiers are trained independently and then combined together via interpolation. Overall, the paper is well organized and easy to follow.

Reasons to accept the paper:
1. The rewritten question oriented verifier could improve the linguistic correctness of the extracted answers.
2. The HAN-based verifier considers the entire document instead of each segment independently, which may enable general transformer-based models to handle long-text document.

Reasons to reject the paper:
1. Some important baselines are not included in the experimental analysis, such as GPT-3 Few-Shot [1] on TriviaQA which achieves 71.2, and RAG [2] on TriviaQA which achieves 68.0.
2. Some important details are missing in the experimental analysis. For example, in Table 2, it is not clear what "DA Verifier" means, and which verifier is used in the method "ALBERT + verifier".
3. It is not clearly discussed the additional computational time and cost spent to train the two proposed verifiers, compared to the baseline without verifiers. For a new method that has marginal performance gain, the extra computational cost should be considered.
4. The illustration of HAN-based verifier in Fig. 1(b) is not complete, which should have included the part for answer prediction and verification loss, etc.


References

[1] Brown, T. B., Mann, B., Ryder, N., Subbiah, M., Kaplan, J., Dhariwal, P., ... & Agarwal, S. (2020). Language models are few-shot learners. arXiv preprint arXiv:2005.14165.

[2] Lewis, P., Perez, E., Piktus, A., Petroni, F., Karpukhin, V., Goyal, N., ... & Riedel, S. (2020). Retrieval-augmented generation for knowledge-intensive nlp tasks. arXiv preprint arXiv:2005.11401.

---

> ### Author Response · Authors · 2020-11-17
> **Express our appreciation for your reviewing and your comments**
>
> Express our appreciation for your reviewing and your comments.
> 1. For the TriviaQA dataset, please also refer to our answer 4 for “AnonReviewer3”. Allow we rewrite some sentences here. After receiving this insightful question on more stronger baselines, especially the GPT-3, T5, and the RAG: “Retrieval-Augmented Generation for Knowledge-Intensive NLP Tasks” (https://arxiv.org/pdf/2005.11401.pdf) (2020/May/22), in its Table 1, TriviaQA results were 56.1/68.0. Another is GPT-3 (https://arxiv.org/pdf/2005.14165v4.pdf), in its Table 3 (F1 scores), we can see that our results are better than that of T5 and GPT-3 Zero-Shot, yet worse than RAG (68.0) or GPT-3 one-shot/few-shot (68.0/71.2). For sure, we will include these baselines in a future submission of this paper. Due to the fact that T5 (and C4 dataset used in T5), BART-Large in RAG together with the whole Wikipedia as a reference, and GPT-3 used extremely larger data/parameters than ALBERT-xxlarge that we used as pre-trained models, these comparisons also reflects that our proposed models on TriviaQA are comparable to some of them.
> Thus, we cautiously appeal that a direct comparison with GPT-3 with 175 billion parameters on 570GB datasets +10-million-USD level cost is possibly less-fair to us. For the RAG baseline, we notice that it also used the whole Wikipedia as the reference knowledge while TriviaQA’s Wikipedia portion datasets are used to testify the performance on information retrieval as well. Generally, we do agree that RAG and GPT-3 with few shot learning have achieved state-of-the-art results on TriviaQA. We would like to include them as reference baselines.
> 2. Actually, by checking URL of the leaderboard, https://rajpurkar.github.io/SQuAD-explorer/, some information is also missing there which caused the ambiguity: (1) “ALBERT+DA Verifier” stands for EM=87.847 (87.8 listed in this paper) and F1=91.265 (91.3 listed in this paper). The group is from “CloudWalk” and all the information we know is its name “ALBERT+Entailment DA Verifier (single model)”. (2) “ALBERT+verifier” stands for EM=88.434 (88.4 listed in this paper) and F1=90.918 (91.0 listed in this paper), when we submit this paper, its name was simply “ALBERT+verifier” by the “QIANXIN” group, currently its name is “aanet_v2.0 (single model)”. (3) SA-NET on Albert” stands for top-1’s “SA-NET on Albert (ensemble)” with EM=90.724 (90.7 listed in this paper) and F1=93.011 (93.0 listed in this paper) from group “QIANXIN”. The name is all we know since no attached papers for them. (4) Retro-Reader online (Zhang et al., 2020b) for “Retro-Reader (ensemble)” from “Shanghai Jiao Tong University” with EM=90.578 (90.6 listed in this paper) and F1=92.978 (93.0 listed in this paper).
> 3. Thanks for the insightful question and we do agree that time comparison for both training and testing should be compared. For comparing with outside baselines, one difficult is that due to the different usage of GPU/TPU hardware and the total number of parameters, we can actually hardly find comparable time-costing information from them, or the time-cost/power-cost such as GPT-3’s reported numbers of month-level training with thousands of NVIDIA V100 cards. For comparing with in-house baselines, such as with ALBERT models without these two verifiers, we would like to briefly share it (even we know after the first submission, these scores need not be taken into account for evaluating the paper): for the first question-rewriting verifier, since we used an external POS-tagger and additionally computed three losses, the time cost for pre-processing the SQuAD2.0, newsQA and TriviaQA cost us one-day around which is totally not necessary in the baselines. After data preprocessing, the training time increased generally and averagely from 95 hours to 124 hours (+30.5%) under a NVIDIA V100 32GB GPU card. The training was mainly for fine-tuning the parameters in ALBERT and in the verifiers. The testing on the dev/test sets also increased at around 22.2% of from 45 minutes to 55 minutes. For the second HAN verifier, the major additional costs are the recurrent layers of sentence level sequences and segment level sequences. Due to the fact that RNN models are difficult to be trained in a non-auto-regression way, the time cost averagely was increased to 160 hours around (+68.4%) for training/fine-tuning and to 83 minutes for testing (+84.4%).
> 4. Thanks for the comment. “answer prediction and verification loss” in Figure 1(b), after H, is actually the same (or part of)  with that described in Figure 1(a). Notice that in Figure 1(a), there is a “H tensor (batch, |q|+|p|+3, h)” layer, actually Figure 1(b)’s not drawn part is same with that in Figure 1(a)’s “predicted answer span” and “answerability” together with span loss and classification loss, respectively. In order to make the figures to be compact, we did not draw them explicitly in Figure 1(b) and for sure we can add them up in a future submission of this paper.

---

### Official Review · AnonReviewer3 · 2020-10-30
**New modules with strong empirical results on MRC**

**Rating:** 7
**Confidence:** 5

**Review:**

In this paper, two linguistic verifiers are proposed to improve the model performance on machine reading comprehension datasets, such as SQuAD v2, NewsQA and TriviaQA. The first verifier rewrites the question by replacing its interrogatives with the predicted answer phrases. Then it computes a score between the rewritten question and the context, so that the answer candidates are position-sensitive. The second verifier leverages a hierarchical attention network, so that the long context can be split in to shorter segments, which are then recurrently connected to conduct answerability classification and boundary determination.

The Empirical results of this proposed method is very strong. Apparently, it achieves a new state-of-the-art performance on the dev set of SQuAD v2. It also outperforms the a bunch of strong baseline methods on the NewsQA dataset. Finally, the proposed model also exceeds the BERT model on TriviaQA.

Overall, it is a good paper.

However, I have some comments:

1. In table 2, what does “Regular Track” mean?

2. In your tables, could you separate the ensemble methods and the single models? It would be much easier to draw a fair comparison.

3. Are you results achieved by ensemble or a single model?

4. You used Albert-xxlarge, but some methods in the tables used smaller pretrained models. For example, in TriviaQA, your baseline is BERT-Large. It might be a bit hard to tell if the improvement is obtained by a better CLM or the proposed modules. So could you do a fair ablation study to verify this?

5. The font of figure 1 looks weird and a bit ugly. I suggest the authors make it more reader friendly.

6. Will you submit your model to the SQuAD v2 leaderboard? I am very interested in seeing its performance on the test set. And I am willing to raise my score if the result aligns with that of dev set (I expect it will top the leaderboard).

***********************************
Post rebuttal: The author has addressed most of my questions, and the SQuAD v2 test result is on par with the state-of-the-art, partially indicating the proposed method is effective. So I am happy to increase my rating and champion for the acceptance.

---

> ### Author Response · Authors · 2020-11-17
> **Appreciate for your insightful comments and questions**
>
> Sincerely appreciate your time and your comments and deeply sorry for a late reply. Here are the answers to these precious questions/comments:
> 1. In table 2, “Regular Track” stands for the category of the results reported from the reference papers. On the other hand, we also list the results from “Top results on the leaderboard” (https://rajpurkar.github.io/SQuAD-explorer/) here since (1) not every result reported in the leaderboard has an attached paper and (2) not every reference paper has submitted their results directly to the leaderboard, and (3) some papers have different results in their paper compared with the results listed in leaderboard (such as the Retro-Reader (Zhang et al., 2020b) reported the best EM/F1 of 88.1/91.4 in their paper while 90.6/93.0 in the leaderboard). In order to distinguish the results, we here separately list them: “Regular Track” for papers’ reported results and “Top results on the leaderboard” for the results submitted to leaderboard. Specially, when the results in leaderboard align with their papers’ results, we also include these baselines in “top results on the leaderboard”.
> 2. In Table 2, “Question-rewritten verifier” and “HAN verifier” stand for single models, and “Combination” stands for ensemble model by employing these two single models and using an ensemble method by referring to (El-Geish, 2020). In addition, quite same with Table 3, Table 4, Table 5, and Table 6: only “Combination” rows are ensemble models (i.e., ensemble of the two verifier models), and other rows (of our results) are single models.
> 3. Please also refer to the answer to Question 2: in Tables 2,3,4,5,6, only “combination” rows are ensemble models’ results and other rows (of our results) are single models’ results.
> 4. Appreciate for the insightful question! We just noticed that for SQuAD2.0 and NewsQA, we included ALBERT-xxlarge enhanced baselines, yet for the TriviaQA dataset, the best baselines that we listed here are only BERT-large which makes the result less comparable. After receiving this insightful question, we further investigated two strong baseline papers, one is “Retrieval-Augmented Generation for Knowledge-Intensive NLP Tasks” (https://arxiv.org/pdf/2005.11401.pdf) (open at 2020/May/22), in its Table 1, the TQA=TriviaQA results were 56.1/68.0. Another is the famous GPT-3 (https://arxiv.org/pdf/2005.14165v4.pdf), in its Table 3 (F1 scores), we can see that our results are better than that of T5 and GPT-3 Zero-Shot, yet worse than RAG (68.0) or GPT-3 one-shot/few-shot (68.0/71.2). We will include these baselines in a future version of this paper. Due to the fact that T5 and GPT-3 used extremely larger data than ALBERT-xxlarge, these comparisons also reflects that our proposed models on TriviaQA are comparable to some of them.
> In addition, we tried hard and could not find a published baseline with ALBERT-xxlarge exactly on TriviaQA. Even we know that new results should not be taken into consideration for evaluating this paper, we still run the experiments of re-implementing the baselines (not new results of our systems) necessary for fair-comparison. We thus run by ourselves two experiments: (1) Google’s implementation of ALBERT-xxlarge-v2 with TriviaQA and (2) the Retro-Reader (Zhang et al.,2020b) which has exactly the same ALBERT-xxlarge-v2 on the TriviaQA wiki datasets. The results for (1) are EM/F1 (%) of 59.2/64.8 which are comparable to our “Question-rewritten verifier”, and (2) Retro-Reader achieved EM/F1 (%) of 60.3/64.8 which are comparable to our individual “HAN verifier”. Based on these, we still would like to argue the HAN verifier is meaningful and significantly better (p<0.05, tested using the same method in Retro-Reader (Zhang et al., 2020)) than the dry-run of ALBERT-xxlarge-v2 and the “combination” was even better. For sure, we will report these results in a future submission of this paper.
> 5. Sorry to say Figure 1 is true ugly and we will try to rewrite by using different fonts and colors.
> 6. Yes, and for sure and we wish to not let you down (and, sorry to say, we could not beat the top-1"SA-NET on Albert (ensemble) by QIANXIN", even quite close. Our results were scheduled to be added to the leaderboard after the anonymity period -- even we are still retraining/fine-tuning it -- hopefully it will be better. For detailed EM/F1 numbers please kindly refer to our comment 2 to "AnnoReviewer 1"). In addition, we are cleaning the code and will release it in a near future. [for sure, we do understand the reviewing policy and these results may not necessary be taken into consideration. Just for reference.]

---

### Decision · Program_Chairs · 2021-01-07
**Final Decision**

**Decision:**

Reject

**Comment:**

The authors propose two linguistic verifiers for improving extractive question answering when the question is answerable. The first replaces the interrogative in the question with candidate answers and evaluates the result both in isolation and in combination with the answer-containing sentence to do answer verification. The second jointly encodes individual sentences and spans with questions in a hierarchical manner to improve use of context in answer prediction performance.

The reviews for this paper are roughly on the cusp: 2 reviewers rate the paper a bit below the acceptance threshold, 1 a bit above, and then 1 now rates the paper as a solid Accept.

Pros

- The main strength of the paper, certainly as emphasized by the most positive reviewer is the strong empirical results. Especially on SQuAD v2, the method here seems to roughly equal the current leading system on the leaderboard.
- The paper also proposes two methods for improving question answering that make sense, are relatively simple, and work

Cons
- The writing and presentation of the paper is not that great. Even at the level of the introduction, the writing just is not very focused: The first page has a lot of background and tutorial information on MRC that just doesn't get to the point of where this paper is situated and what it contributes.
- Neither of the proposed systems are that novel (though it is interesting to see that they still have value even in the age of large contextual language models)
- The paper lacks ML novelty
- The methods appear to be significantly more expensive to run
- Some empirical comparisons appear to be lacking

As well as the missing comparisons mentioned by some reviewers, I think that there are a number of other missing relevant datapoints. While not denying that gathering the available results for NewsQA/TriviaQA is much less straightforward than with that nice leaderboard for SQuAD, aren't there are lot of systems with better results on TriviaQA that aren't mentioned in the paper. These include: RoBERTa and SpanBERT (mandarjoshi); BigBird-ETC see https://proceedings.neurips.cc/paper/2020/file/c8512d142a2d849725f31a9a7a361ab9-Paper.pdf; Longformer; SLQA see https://www.aclweb.org/anthology/P18-1158.pdf .

But, overall, I think the decision on this paper comes down to focus and contributions. Not withstanding the growing size of ICLR, I would like to think that it is not just another ML and ML applications conference, but it is a conference centered on representation learning. The present paper, no matter its quality and strong results, just isn't a contribution to representation learning. It is a much better fit to an NLP conference where it would be a strong contribution to question answering, showing the continuing value of linguistic methods like question rewriting in answer validation. But this just isn't a contribution within the focus of representation learning. Just as R4 does, I encourage the authors to clean up the presentation of the paper a bit and to submit it to an NLP conference, where it would be a strong contribution, for the reasons that R3 emphasizes.